# Favorable Impact of a Multidisciplinary Team Approach on Heart Transplantation Outcomes in a Mid-Volume Center

**DOI:** 10.3390/jcm11092296

**Published:** 2022-04-20

**Authors:** Jun Ho Lee, Joo Yeon Kim, Ilkun Park, Kiick Sung, Wook Sung Kim, Darae Kim, Jeong Hoon Yang, Eun-Seok Jeon, Jin-Oh Choi, Nayeon Choi, Hanpyo Hong, Yang Hyun Cho

**Affiliations:** 1Department of Thoracic and Cardiovascular Surgery, Samsung Medical Center, Sungkyunkwan University School of Medicine, Seoul 06351, Korea; ecmo1984@gmail.com (J.H.L.); thestyner@hotmail.com (I.P.); kiick.sung@samsung.com (K.S.); wooksung.kim@samsung.com (W.S.K.); 2Department of Thoracic and Cardiovascular Surgery, Incheon Sejong Hospital, Incheon 21080, Korea; jydream85@naver.com; 3Department of Cardiology, Samsung Medical Center, Sungkyunkwan University School of Medicine, Seoul 06351, Korea; darae0918.kim@samsung.com (D.K.); jhysmc@gmail.com (J.H.Y.); eunseok.jeon@samsung.com (E.-S.J.); choijean5@gmail.com (J.-O.C.); 4Biostatistical Consulting and Research Lab, Medical Research Collaborating Center, Hanyang University, Seoul 04763, Korea; nayeon@hanyang.ac.kr; 5Artificial Intelligence & Big Data Medical Center, Life Log Bigdata Platform Business Group, Yonsei University, Wonju College of Medicine, Wonju 26493, Korea; drred9240@yonsei.ac.kr

**Keywords:** multidisciplinary team, critical care, heart failure, heart transplantation, online information sharing

## Abstract

Although a multidisciplinary team (MDT) approach is recommended for advanced heart failure and heart transplantation (HTx), no studies have investigated the impact of the team approach on post-HTx survival. Thus, we implemented an MDT approach in our HTx program in 2014, with the active involvement of critical care and extracorporeal life support (ECLS) teams and the use of a real-time online information sharing system. We hypothesized that this MDT approach would result in improved survival of patients who had undergone HTx. We enrolled 250 adult patients who underwent HTx between December 2003 and June 2018. They were divided into non-MDT (*n* = 120; before 2014) and MDT (*n* = 130; since 2014) groups. The primary outcome was overall mortality. In terms of donor age, diabetes, dialysis, ECLS, and waiting time, the MDT group had more high-risk patients. The MDT approach was found to be an independent predictor of overall survival using a variety of multivariable analytic methods, including inverse probability of treatment weighting analysis. An HF team, a critical care team, and an ECLS team collaboration may improve survival following HTx. To improve the efficiency of the MDT approach, we recommend using a real-time online information sharing system.

## 1. Introduction

The guideline of the American College of Cardiology Foundation and the American Heart Association strongly recommends the use of the team approach for treating heart failure (HF) [1]. As per the International Society for Heart and Lung Transplantation, a multidisciplinary team (MDT) should include nearly all teams and healthcare workers related to the heart transplantation (HTx) procedure [2]. Several studies have reported that a team approach is considered the gold standard model for treating patients with HF [3,4,5,6,7,8,9,10,11]. Although many HF centers have developed and operated their own HF teams [6,8], no studies have yet examined on the impact of a team approach, such as on the survival of patients following HTx. In the last decade, there has been an increasing need for HF centers to perform high-risk transplantations such as HTx for recipients in intensive care units (ICUs) because of the need for mechanical life support [12,13,14,15]. We believe that effective collaboration with intensive care and extracorporeal life support (ECLS) teams can improve high-risk HTx outcomes.

Communication among many team members from various departments, teams, and centers, on the other hand, has been identified as a major concern. To address this, the HF team has implemented a real-time online information sharing system based on a social network service (SNS). The influence of our MDT approach on key clinical outcomes, such as patient survival following HTx, was then investigated.

## 2. Methods

### 2.1. MDT Approach

In 2014, our HF team began working closely with the center for intensive care, which includes cardiac, pulmonary, and surgical intensive care specialists and an ECLS team. Prior to 2014, it was not a true multidisciplinary approach; rather, there were periodic formal consultations between clinicians who needed patient information. It used to be a lot more passive and non-systematic. As the HF team expanded, it became vital to implement an online information sharing tool to improve the MDT approach’s efficacy. Although MDTs frequently used person-to-person discussions through physical meetings and phone calls, we also decided to use a secure online group chat room provided by an SNS. In the online chat room for patients with advanced HF and cardiogenic shock, approximately 15 key people were added, including HF cardiologists, a chief cardiac intensivist, the surgical director of HTx, HF nurses, ECLS specialists, a cardiologist, and the cardiac surgeon on in-house duty. In the online group chat room, anyone can raise concerns about any patient who requires the attention of the HF team (Figure A1). The MDT approach was defined as a collaboration between a traditional HF team, an intensive care team, and an ECLS team that communicated through a real-time online information sharing system (available 24 h a day, 7 days a week).

Our MDT approach has been applied to all patients with acute HF, including cardiogenic shock, and those with chronic HF. All critical professionals involved in acute and chronic HF and cardiogenic shock communicate significant information (e.g., a new referral from outside) and any substantial changes in pre-existing patients (e.g., cardiogenic shock in a HTx candidate on the floor) because the MDT includes HF/HTx physicians, intensivists, an ECLS team, and in-house cardiovascular surgeons. Through the online group chat room, all members of the MDT could access the patient’s information and express their opinions, and the MDT made it possible to share information more efficiently and effectively than through physical meetings and phone calls. In Table A1, the features of an MDT are compared to those of a non-MDT. 

### 2.2. Study Population, Data Collection, and Endpoints

In total, 259 adult patients underwent HTx at Samsung Medical Center from December 2003 to June 2018 (Figure 1). All patients were on the waiting list for HTx in the Korean Network for Organ Sharing. We excluded nine patients who underwent repeated HTx. In the final cohort of 250 patients, 130 (52.0%) were treated and monitored postoperatively utilizing an MDT approach (MDT group), and those who had HTx before 2014 were assigned to the non-MDT group.

The primary outcome of this study was overall mortality. The baseline characteristics of the clinical and laboratory data of the patients were collected from electronic medical records and databases. A trained coordinator collected follow-up clinical data, including vital status, by reviewing the medical records and telephone interviews. The National Registry of Births and Deaths confirmed this information by utilizing a unique personal identification number for each patient to complete mortality data.

### 2.3. Statistical Analysis

The baseline and clinical characteristics of the study population are presented as mean ± standard deviation, median with interquartile range, or frequency and proportion. Wilcoxon rank-sum tests were used to compare skewed continuous variables. The chi-square test was used to compare categorical variables between the two groups. The Cox proportional hazards regression model for univariable and multivariable analyses and the forest plot based on the results of the multivariable analysis were used to determine the independent predictors of overall mortality. Multivariable analyses were performed on the significant variables in the univariable analyses. To estimate the survival curves during the follow-up period, we used the Kaplan-Meier method, whereas the log-rank test was used to compare the survival rates between the two groups.

We then adjusted for the differences in baseline characteristics using weighted Cox proportional hazards regression models with inverse probability of treatment weighting (IPTW) to reduce potential confounding factors [16]. The adjusted variables are summarized in Table A2. For the continuous and dichotomous variables, the standardized difference was determined using the mean and prevalence, respectively (Table 1).

All tests were two-tailed. Statistical significance was defined as a *p* value of less than 0.05. R software (version 3.5.1; R Foundation for Statistical Computing, Vienna, Austria) and SAS software (version 9.4; SAS Institute Inc., Cary, NC, USA) were used for statistical analysis.

### 2.4. Ethics

This study was approved by the institutional review board of Samsung Medical Center (Seoul, Korea) after waiving the need for informed consent (IRB number 2020-09-115).

## 3. Results

### 3.1. Baseline Characteristics

The mean age of the HTx recipients was 51.0 ± 13.7 years, and the median age was 54 years (range: 18–78 years). Of the 250 patients, 77 (30.8%) were female. Patients in the MDT group were more likely to have diabetes, continuous renal replacement therapy, a history of percutaneous coronary intervention, a longer waiting time for HTx, and more frequent use of pre-HTx ECLS than those in the non-MDT group. The mean duration of ECLS before HTx was 15.2 ± 18.9 days, and the median duration was 10.0 days (range: 1–126 days, interquartile range: 6–17 days). The frequency of pre-HTx ECLS was noted to be significantly different between the two groups (non-MDT group: 23.3% vs. MDT group: 49.2%; *p* < 0.001). Significant inter-group differences were also observed in the baseline covariates of hypertension, previous cardiac surgery, dialysis, HF with preserved ejection fraction, ischemic cardiomyopathy, and the frequency of pre-HTx ECLS after adjusting for the baseline profiles using IPTW. The baseline characteristics of the patients in the non-MDT and MDT groups are summarized in Table 1.

### 3.2. Operative and Postoperative Characteristics

Although cold ischemic time was longer for the MDT group than for the non-MDT group, the total ischemic times were similar due to the shorter warm ischemic time and aortic cross-clamping time in the MDT group. The post-HTx ICU stay and total ICU stay were longer for the MDT group than for the non-MDT group, but the post-HTx hospital stay times were similar (*p* = 0.993). The frequency of post-HTx ECLS did not differ between the two groups (non-MDT group: 10.0% vs. MDT group: 13.8%; *p* = 0.350). No significant difference was observed as regards the frequency of new ECLS after HTx between both groups (*p* = 0.069). The operative and postoperative data are summarized in Table 2. The immunosuppression strategy at three months after HTx is described in Figure 2. Tacrolimus and mycophenolate were commonly used during the study period, whereas everolimus was used only in the MDT group.

### 3.3. Impact of the MDT and Predictors of Overall Mortality

The Kaplan–Meier survival analysis revealed an inter-group difference in the overall survival after HTx (log-rank *p* = 0.012) (Figure 3). Additionally, the cardiac-related survival after HTx was determined to be different in the two groups (log-rank *p* = 0.012) (Figure A2). The MDT approach was an independent predictor of survival (*p* = 0.001; hazard ratio: 0.341; 95% confidence interval: 0.182–0.637) in the multivariable analysis before IPTW for overall mortality. The age of HTx recipients (analyzed 10-year increases; *p* = 0.002; hazard ratio: 1.371; 95% confidence interval: 1.128–1.666) and the level of total bilirubin (*p* = 0.024; hazard ratio: 1.040; 95% confidence interval: 1.005–1.076) were observed to be independent predictors of death. The results of the Cox proportional hazards regression model for univariable and multivariable analyses of overall mortality are summarized in Figure 4A and Table A3.

The MDT approach was found to be an independent predictor of survival (*p* < 0.001; hazard ratio: 0.312; 95% confidence interval: 0.198–0.491) after adjustment using IPTW. Additionally, the age of the HTx recipients (analyzed 10-year increases; *p* < 0.001; hazard ratio: 1.408; 95% confidence interval: 1.207–1.642) and the level of total bilirubin (*p* < 0.001; hazard ratio: 1.050; 95% confidence interval: 1.022–1.078) were also found to be independent predictors of death after adjustment using IPTW (Figure 4B and Table A3, right columns). In contrast, the duration of waiting for HTx was an independent predictor of survival (*p* = 0.011; hazard ratio: 0.982; 95% confidence interval: 0.968–0.996). The adjusted outcomes of various statistical methods are summarized in Table 3. The MDT group had a lower risk of overall mortality compared to the non-MDT group (*p* < 0.001) when the adjustment using IPTW was further augmented by the multivariable analyses (IPTW + multivariable analyses).

### 3.4. Early Clinical Outcomes

In all patients, the 30-day and 1-year mortality rates were 4.4% (*n* = 11) and 13.6% (*n* = 34), respectively. The most common causes of 1-year mortality were septic shock (11/34, 32.3%), rejection (8/34, 23.5%), and sudden cardiac death, including unknown causes (8/34, 23.5%). No significant difference was determined in terms of the 30-day mortality rate between both groups (non-MDT group: 5.8% vs. MDT group: 3.1%; *p* = 0.288). However, the 1-year mortality rate was significantly higher in the non-MDT group (non-MDT group: 18.3% vs. MDT group: 9.2%; *p* = 0.036). A difference in 1-year survival following HTx was also determined using Kaplan–Meier survival analysis (log-rank *p* = 0.040) (Figure A3).

## 4. Discussion

In this study, we were able to determine the following: (a) the MDT approach may improve patient outcomes after HTx, (b) the MDT group used more pre-HTx ECLS, (c) the MDT group had more high-risk HTx patients, and (d) both surgical and medical management changed significantly after using the MDT approach. Despite the fact that most large HTx programs already use the team approach in their HF programs [3,4,5,6,7,8,9,10,11], no studies have looked at its impact on patient survival after HTx. Although one review focused on changes in the quality of care provided after implementing the team approach in their HTx program, they found no significant differences in terms of survival or pre/post-HTx management [17]. Aside from forming an HF team, we feel it is also critical to understand how to effectively operate an HF team. A large-scale MDT, including a critical care team and an ECLS team, has been developed. We also used a secure online chat room provided by an SNS, which can be a tool to improve the efficiency of our MDT.

In our results, the baseline covariates of hypertension, previous cardiac surgery, dialysis, ischemic cardiomyopathy, and the use of pre-HTx ECLS were significantly higher in the MDT group than in the non-MDT group after adjusting for baseline profiles using IPTW. Moreover, the use of pre-HTx ECLS had a significant inter-group difference between the two groups both before (*p* < 0.001) and after (*p* = 0.006) adjustment using IPTW (Table 1). Therefore, HTx progressed in recipients; the MDT group was at greater risk. Furthermore, the Sequential Organ Failure Assessment (SOFA) score tended to be higher in the MDT group as well. Since patients in the MDT group had worse preoperative conditions, the duration of the post-HTx ICU stay was noted to be significantly longer, and the frequencies of post-HTx ECLS and new post-HTx ECLS tended to be higher as well. Nevertheless, the 1-year, cardiac-related, and overall survival rates in the MDT group were significantly better than those in the non-MDT group, suggesting that the post-HTx care was well performed through the MDT approach. The favorable response could be explained by the MDT approach’s positive effect of including a cardiac and general critical care team and an ECLS team [18]. 

In this study, we aimed to use the MDT approach to overcome the increasing number of high-risk HTx patients. We collaborated with a critical care team and an ECLS team. The majority of the significant changes in the patients’ conditions were immediately shared in our secure online chat room. We also tried deploying the ECLS system before the occurrence of severe end-organ damage or sudden cardiac arrest. Interventional or surgical left heart decompression and distal limb protection procedures were performed prophylactically or during an exceedingly early stage of ECLS. The principles of modern critical care, including light or no sedation, adequate pain control, and active rehabilitation, among others, were applied to all patients in our service. 

As the real-time online information sharing system is available 24 h a day, 7 days a week, some members who are off-duty or on vacation may be disturbed by frequent alert messages. Although the person off-duty can turn off the message alerts, it is inevitable that he will check the message’s contents. Staff physicians and surgeons participate in the chat room on their mobile phones. At the same time, other lower-level doctors and nurses generally use their on-call phones in the secure online group chat room to minimize such problems. The real-time online information sharing system could be very effective to immediately share a variety of opinions that all team members should be aware of. However, it is prudent to reach the conclusion that the secure online chat room should completely replace the previous offline multidisciplinary treatment. Major clinical decisions were still communicated and discussed over the phone or in person. We are currently looking for a new application that will maintain both members’ privacy and efficient information sharing while keeping all members on the same level of recognition of HF patients.

Additionally, prioritizing a patient for ECLS or other temporary mechanical circulatory support methods was an important approach for our team. Donors meeting the extended criteria were accepted to reduce the waiting time of HTx recipients. Donors older than 50 years or expecting a cold ischemic time of up to four hours were accepted for HTx patients on temporary mechanical circulatory support. This was done by modifying the surgical technique to decrease the warm ischemic time. Postoperative medical management, including infection prevention and control, immune suppression, and monitoring protocols for acute and chronic rejection, were all revised. The postoperative concerns of HTx patients were also discussed at regular physical meetings between physicians and surgeons, and in a secure online chat room.

There was no significant difference in the 30-day mortality rate between both groups, even though the MDT group had more high-risk patients. With the MDT approach, we have wisely used organ-support devices such as ECLS and continuous renal replacement therapy to protect end-organ functions. We have also tried to reduce unnecessary waiting times on those devices by accepting extended-criteria donors. After experiencing similar or improved outcomes, we have become comfortable in handling challenging high-risk HTx patients with the MDT approach. In our center, the MDT has been applied in other critical treatment areas with a disease-focused approach. This real-time online information sharing system has also been introduced. The MDT approach has resulted in improved clinical outcomes in ECLS at our institution [19,20]. 

### Study Limitations

We also recognize several limitations of this study. First, because this was a single-center study with 250 patients, the findings in our cohort may not be generalizable to all HTx patients. Second, because this was a retrospective observational study comparing two periods, there may have been residual confounding factors after statistical adjustments. Aside from the effect of the MDT approach, the outcomes may have also been influenced by the cumulative experience in performing surgical techniques and perioperative management. These changes in the MDT group were deemed unavoidable, and so we performed various adjustments to control for confounders and to reduce this limitation, including IPTW and regression modeling using several risk factors. According to our results, the MDT group had more high-risk patients, including increased use of pre-HTx ECLS. However, the MDT approach showed improvements in the outcomes of patients after HTx. We investigated 1-year survival for each year, and the gap in outcomes had a significant change around 2014. Additionally, since 2014, the 1-year survival rate has never gone below 80% (Figure A4). The difference in 1-year survival between the MDT and the non-MDT groups showed that the MDT approach was a more important factor, which then significantly affected the outcomes more than the cumulative magnitude of benefit over time, such as accumulated experience in performing surgical techniques and perioperative management. Third, the duration of ECLS before HTx was shorter in this study (median: 10.0 days, mean: 15.2 ± 18.9 days) than in some reports with ECLS in the United States [12,13] but was comparable or longer compared to other previous publications [14,15]. Fourth, despite the fact that the immunosuppression strategy at three months after HTx was depicted in Figure 2, performing a univariable and multivariable analysis of the immunosuppressive medication therapy was technically difficult. This is due to the fact that not only one immunosuppressive drug was utilized per patient, but it was also administered to a number of patients as part of a combination therapy.

## 5. Conclusions

The MDT approach was found to be a significant predictor of overall survival. An HF team, a critical care team, and an ECLS team collaboration may improve survival following HTx. To improve the efficiency of the MDT approach, we recommend using a real-time online information sharing system.

## Figures and Tables

**Figure 1 jcm-11-02296-f001:**
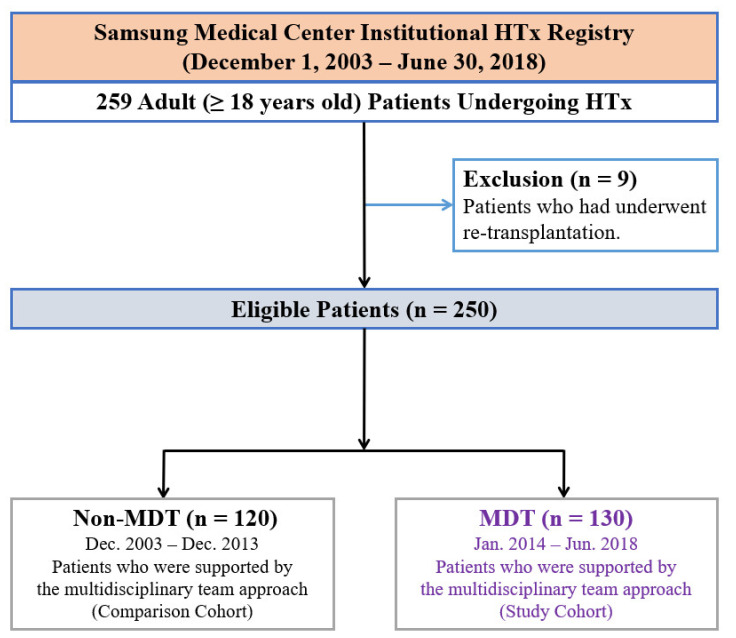
Flow diagram of patient recruitment. HTx, heart transplantation; MDT, multidisciplinary team.

**Figure 2 jcm-11-02296-f002:**
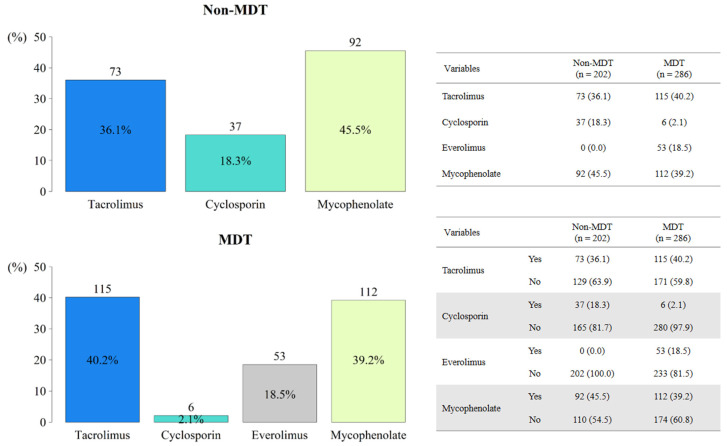
Use of immunosuppressive agents at three months after heart transplantation. MDT, multidisciplinary team.

**Figure 3 jcm-11-02296-f003:**
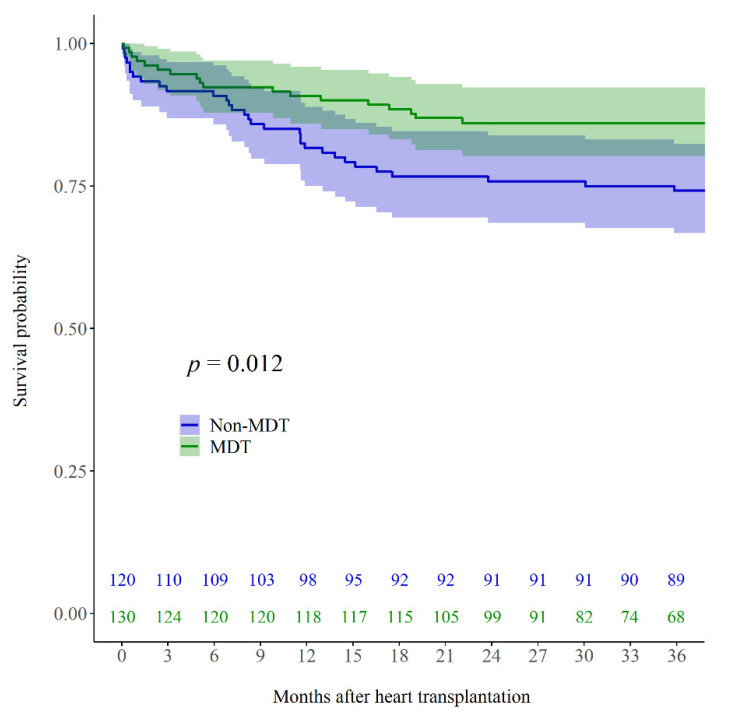
Kaplan–Meier post-heart transplantation survival curves for overall survival in patients who underwent heart transplantation with the multidisciplinary team approach (MDT group; green line) and without the multidisciplinary team approach (non-MDT group; blue line). MDT, multidisciplinary team.

**Figure 4 jcm-11-02296-f004:**
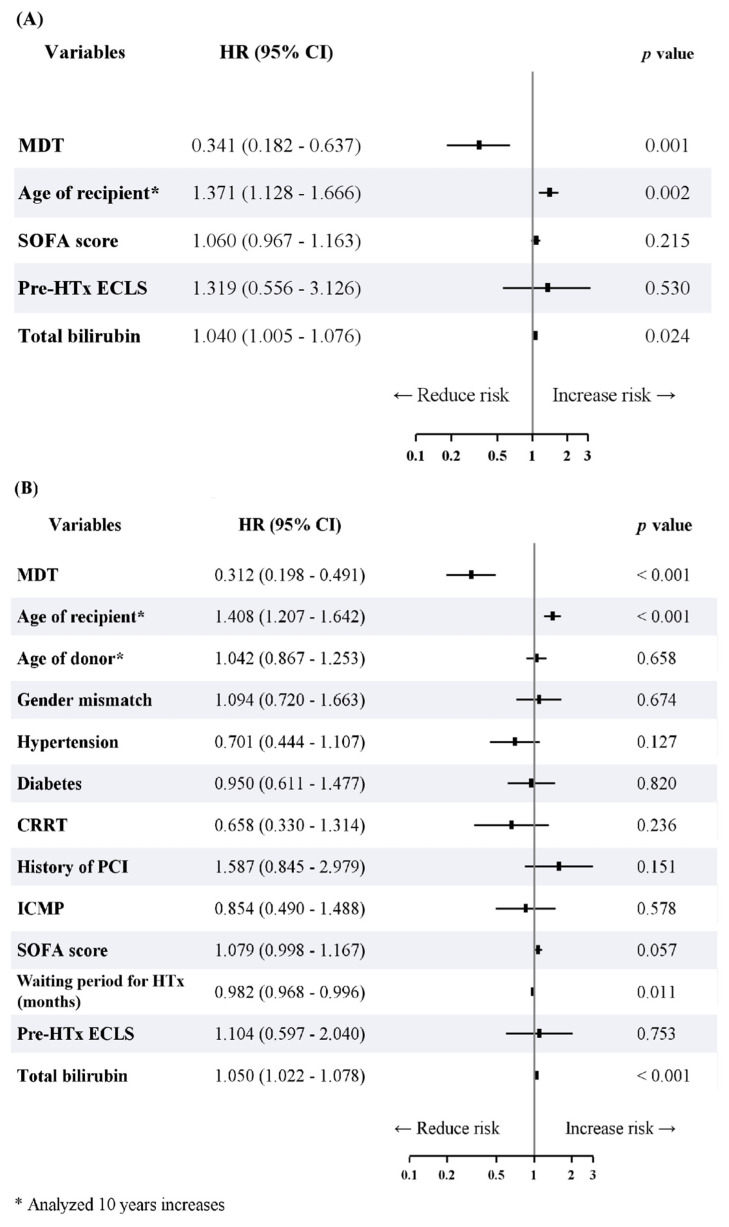
Forest plot based on the results of multivariable analysis of overall mortality (**A**) before adjustment using IPTW and (**B**) after adjustment using IPTW. IPTW, inverse probability of treatment weighting; HR, hazard ratio; CI, confidence interval; MDT, multidisciplinary team; SOFA, Sequential Organ Failure Assessment; HTx, heart transplantation; ECLS, extracorporeal life support; CRRT, continuous renal replacement therapy; PCI, percutaneous coronary intervention; ICMP, ischemic cardiomyopathy.

**Table 1 jcm-11-02296-t001:** Baseline characteristics of all patients.

	Overall Cohort					IPTW	
	Total (*n* = 250)	Non-MDT (*n* = 120)	MDT (*n* = 130)	*p* Value	Standardized Difference	*p* Value	Standardized Difference
Age of recipient (years)	54.0 (43.8–61.0)	52.5 (40.3–61.0)	55.0 (45.0–61.0)	0.121	0.217	0.312	0.060
Female recipients	77 (30.8)	41 (34.2)	36 (27.7)	0.268	−0.140	**<0.001**	−0.559
Age of donor (years)	43.0 (34.0–50.0)	41.0 (34.0–48.0)	44.5 (33.8–51.3)	**0.012**	0.271	0.146	0.235
Female donors	64 (25.6)	31 (25.8)	33 (25.4)	0.935	−0.010	**<0.001**	−0.449
Gender mismatch	81 (32.4)	38 (31.7)	43 (33.1)	0.812	0.030	0.204	0.109
Same ABO type matching	191 (78.3)	90 (78.9)	101 (77.7)	0.812	−0.031	0.242	−0.100
Hypertension	88 (35.2)	43 (35.8)	45 (34.6)	0.840	−0.026	**<0.001**	−0.399
Diabetes	56 (22.4)	20 (16.7)	36 (27.7)	**0.037**	0.268	0.160	0.120
Stroke	21 (8.4)	11 (9.2)	10 (7.7)	0.675	−0.053	0.618	0.043
Previous cardiac surgery	52 (20.8)	23 (19.2)	29 (22.3)	0.541	0.078	**<0.001**	0.331
LVAD	14 (5.6)	4 (3.3)	10 (7.7)	0.134	0.192	0.179	0.114
Dialysis	38 (15.3)	16 (13.3)	22 (17.2)	0.400	0.107	**0.006**	0.235
CRRT	30 (12.1)	9 (7.5)	21 (16.4)	**0.032**	0.277	0.091	0.144
History of PCI	43 (17.2)	14 (11.7)	29 (22.3)	**0.026**	0.286	0.238	0.101
LVEF (%)	21.0 (17.0–27.0)	21.5 (17.0–27.8)	21.0 (17.0–27.0)	0.613	−0.122	0.953	0.088
HFpEF	28 (11.2)	16 (13.3)	12 (9.2)	0.304	−0.130	**0.007**	0.230
ICMP	56 (22.4)	21 (17.5)	35 (26.9)	0.074	0.228	**0.032**	0.183
SOFA score	7.0 (4.0–11.0)	6.0 (4.0–10.0)	8.0 (5.0–12.0)	0.052	0.244	0.087	0.258
Waiting period for HTx (days)	49.0 (18.0–124.5)	36.0 (14.3–80.8)	72.5 (22.0–167.0)	**0.002**	0.300	0.683	−0.449
Pre-HTx ECLS	92 (36.8)	28 (23.3)	64 (49.2)	**<0.001**	0.559	**0.006**	0.234
Pre-HTx ECLS duration (days)	10.0 (6.0–17.0)	8.0 (2.0–14.5)	11.0 (7.0–18.8)	0.050	0.394	0.101	0.270
Hemoglobin (g/dL)	10.8 (9.6–12.5)	11.3 (9.7–12.9)	10.5 (9.4–12.1)	**0.019**	−0.308	**0.044**	−0.324
Platelet count(×10^3^/mm^3^)	149.0 (100.0–214.0)	158.0 (111.0–216.3)	143.0 (93.3–214.0)	0.157	−0.203	0.265	−0.166
Creatinine (mg/dL)	1.1 (0.9–1.5)	1.2 (0.9–1.5)	1.1 (0.9–1.5)	0.381	0.093	0.504	0.205
Total bilirubin (mg/dL)	1.7 (0.9–2.9)	1.6 (0.9–3.0)	1.7 (0.8–2.8)	0.879	0.120	0.185	0.106
Albumin (g/dL)	3.6 (3.1–4.1)	3.6 (3.0–4.2)	3.6 (3.1–4.1)	0.836	0.007	0.349	−0.181
NT-proBNP (pg/mL)	8058.0 (3799.5–18,433.5)	7792.0 (3813.0–21,543.0)	8105.0 (3742.8–16,024.8)	0.640	−0.149	0.105	−0.137

Non-normally distributed numerical variables are presented as medians (interquartile ranges) and were analyzed using the Wilcoxon rank-sum test. Categorical variables are presented as numbers (percentages) and were analyzed using the chi-square test. IPTW, inverse probability of treatment weighting; MDT, multidisciplinary team; LVAD, left ventricular assist device; CRRT, continuous renal replacement therapy; PCI, percutaneous coronary intervention; LVEF, left ventricular ejection fraction; HFpEF, heart failure with preserved ejection fraction; ICMP, ischemic cardiomyopathy; SOFA, Sequential Organ Failure Assessment; HTx, heart transplantation; ECLS, extracorporeal life support; NT-proBNP, *N*-terminal prohormone of brain natriuretic peptide. Values in bold typeface indicate statistically significant findings.

**Table 2 jcm-11-02296-t002:** Operative and postoperative characteristics of all patients.

	Total (*n* = 250)	Non-MDT (*n* = 120)	MDT (*n* = 130)	*p* Value
CPB time (minutes)	147.0 (126.0–177.0)	159.0 (135.0–190.5)	139.0 (121.0–161.0)	**<0.001**
ACC time (minutes)	86.0 (68.0–105.0)	97.0 (86.0–112.0)	70.0 (60.0–87.0)	**<0.001**
Total ischemic time (minutes)	181.0 (147.0–235.0)	173.0 (143.0–237.0)	187.0 (150.0–235.0)	0.383
Cold ischemic time (minutes)	116.0 (82.0–169.0)	100.0 (70.0–163.0)	120.0 (96.0–174.0)	**0.003**
Warm ischemic time (minutes)	65.0 (54.0–76.0)	70.0 (63.0–82.0)	56.0 (47.0–68.0)	**<0.001**
Post-HTx ECLS	30 (12.0)	12 (10.0)	18 (13.8)	0.350
Post-HTx new ECLS	8 (5.1)	2 (2.2)	6 (9.1)	0.069
Post-HTx ICU stay (days)	10.0 (6.0–16.0)	7.0 (5.0–14.0)	12.0 (8.0–20.0)	**<0.001**
Total ICU stay (days)	14.0 (7.0–28.3)	10.5 (5.3–21.8)	17.0 (9.0–35.0)	**<0.001**
Post-HTx hospital stay (days)	30.0 (23.0–50.0)	30.5 (24.0–41.0)	30.0 (22.0–59.3)	0.993

Non-normally distributed numerical variables are presented as medians (interquartile ranges) and were analyzed using the Wilcoxon rank-sum test. Categorical variables are presented as numbers (percentages) and were analyzed using the chi-square test or Fisher’s exact test. MDT, multidisciplinary team; CPB, cardiopulmonary bypass; ACC, aorta cross-clamping; HTx, heart transplantation; ECLS, extracorporeal life support; ICU, intensive care unit. Values in bold typeface indicate statistically significant findings.

**Table 3 jcm-11-02296-t003:** Adjusted hazard ratios for overall, 1-year, and 3-year mortality following the MDT group compared with the non-MDT group.

Outcomes	*p* Value	Hazard Ratio	95% Confidence Interval	
			Lower 0.95	Upper 0.95
Overall Mortality				
Crude	**0.014**	0.486	0.274	0.863
IPTW	**0.002**	0.506	0.327	0.783
IPTW + multivariable *	**<0.001**	0.396	0.253	0.621
1-year mortality				
Crude	**0.044**	0.485	0.240	0.981
IPTW	**0.007**	0.487	0.290	0.817
IPTW + multivariable *	**0.001**	0.403	0.237	0.687
3-year mortality				
Crude	**0.024**	0.513	0.287	0.918
IPTW	**0.007**	0.542	0.348	0.843
IPTW + multivariable *	**0.001**	0.457	0.290	0.721

* Baseline variables that were significantly different after IPTW: hypertension, previous cardiac surgery, previous dialysis, heart failure with preserved ejection fraction, ischemic cardiomyopathy, and the frequency of pre-HTx ECLS. MDT, multidisciplinary team; IPTW, inverse probability of treatment weighting; HTx, heart transplantation; ECLS, extracorporeal life support. Values in bold typeface indicate statistically significant findings.

## Data Availability

The authors confirm that the data supporting the findings of this study are available within the article and its Appendix A, Appendix B, Appendix C, Appendix D, Appendix E, Appendix F and Appendix G. Raw data that support the findings of this study are available from the corresponding author, upon reasonable request.

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
