# Peer review of "Favorable Impact of a Multidisciplinary Team Approach on Heart Transplantation Outcomes in a Mid-Volume Center"

_jcm, 2022, doi:10.3390/jcm11092296_

Round 1
Reviewer 1 Report
Favorable impact of a multidisciplinary team approach on heart transplantation outcomes in a mid-volume center
Dear Editor, thank you for the opportunity to review this article.
The authors conducted a beautiful work on multidisciplinary approach to patients with heart failure.
The paper is very well written and conducted according to guidelines.
I have nothing to add to further improve the text.
Author Response
Your favorable feedback is much appreciated.
Thank you for your comment.
We sincerely hope that our study will be published in the Journal of Clinical Medicine special issue “Contemporary Management of Patients with Heart Failure.”
We are grateful for your positive reviews.
Thank you very much.
Sincerely,
Jun Ho Lee, MD, PhD (1st author, ecmo1984@gmail.com)
Yang Hyun Cho, MD, PhD (Corresponding author, mdcho95@gmail.com)
Reviewer 2 Report
The paper entitled “Favourable impact of a multidisciplinary team approach on heart transplantation outcomes in a mid-volume center” seems to be very well written. The article runs smoothly. The topic is very actual and interesting, even though MDT approach is an already established reality in the most important transplant centers.
Unsurprisingly, you reported that MDT approach was found to be an independent predictor of survival (p < 0.001; 182 hazard ratio: 0.312; 95% confidence interval: 0.198–0.491) together with the age of the HTx recipients and the level of total bilirubin. How do you explain it? Do you think every morning meeting could lead to the same results? (doi: 10.1111/ctr.14061)
MDT is of paramount importance in the era of personalized medicine (doi: 10.4143/crt.2017.125), but a chat meeting could improve confusion and not shared decisions if not established when the all team is simultaneously available. How did you resolve it?
Interestingly, there was no significant difference in the 30-day mortality rate between both groups (non-MDT group: 5.8% vs. MDT group: 3.1%; p = 0.288) even though the MDT group had a significantly higher-risk patients in terms of age of donors, diabetes, dialysis, ECLS, and waiting time as you reported.
Author Response
Thank you for your thorough and informative remarks. We've considered all of your suggestions, and our responses are provided below.
Comment 1:
The paper entitled “Favourable impact of a multidisciplinary team approach on heart transplantation outcomes in a mid-volume center” seems to be very well written. The article runs smoothly. The topic is very actual and interesting, even though MDT approach is an already established reality in the most important transplant centers.
Answer 1:
Your favorable feedback is much appreciated.
Thank you for your comment.
Comment 2:
Unsurprisingly, you reported that MDT approach was found to be an independent predictor of survival (p < 0.001; 182 hazard ratio: 0.312; 95% confidence interval: 0.198–0.491) together with the age of the HTx recipients and the level of total bilirubin. How do you explain it? Do you think every morning meeting could lead to the same results? (doi: 10.1111/ctr.14061)
Answer 2:
Thank you for your comment.
We reported that the age of the HTx recipients (analyzed 10 years increases; p < 0.001; hazard ratio: 1.408; 95% confidence interval: 1.207–1.642) and the level of total bilirubin (p < 0.001; hazard ratio: 1.050; 95% confidence interval: 1.022–1.078) were found to be independent predictors of death after adjustment using IPTW.
Independent from these factors, the MDT approach was found to be an independent predictor of survival (p < 0.001; hazard ratio: 0.312; 95% confidence interval: 0.198–0.491) after adjustment using IPTW.
The major finding of this study is that the MDT approach is an independent predictor of a variety of characteristics that can significantly reflect the prognosis of HTx recipients.
Comment 3:
MDT is of paramount importance in the era of personalized medicine (doi: 10.4143/crt.2017.125), but a chat meeting could improve confusion and not shared decisions if not established when the all team is simultaneously available. How did you resolve it?
Answer 3:
We sincerely thank you for your keen interest.
The real-time online information sharing system could be very effective to immediately share a variety of opinions that all team members should be aware of. However, it is prudent to reach the conclusion that the secure online chat room can completely replace the previous offline multidisciplinary treatment. Major clinical decisions were still communicated and discussed over the phone or in person.
The Discussion section of the manuscript was revised accordingly (Lines 232-236).
Comment 4:
Interestingly, there was no significant difference in the 30-day mortality rate between both groups (non-MDT group: 5.8% vs. MDT group: 3.1%; p = 0.288) even though the MDT group had a significantly higher-risk patients in terms of age of donors, diabetes, dialysis, ECLS, and waiting time as you reported.
Answer 4:
We sincerely thank you for your keen interest.
There was no significant difference in the 30-day mortality rate between both groups, even though the MDT group had more high-risk patients. With the MDT approach, we have wisely used organ support devices such as ECLS and continuous renal replacement therapy to protect end-organ functions. We have also tried to reduce unnecessary waiting time on those devices by accepting extended-criteria donors. After experiencing similar or improved outcomes, we have become comfortable in handling challenging high-risk HTx patients with the MDT approach.
The Discussion section of the manuscript was revised accordingly (Lines 248-253).
We sincerely hope that our study will be published in the Journal of Clinical Medicine special issue “Contemporary Management of Patients with Heart Failure.”
We are grateful for your positive reviews.
Thank you very much.
Sincerely,
Jun Ho Lee, MD, PhD (1st author, ecmo1984@gmail.com)
Yang Hyun Cho, MD, PhD (Corresponding author, mdcho95@gmail.com)

Reviewer 3 Report
The paper deals with the interesting subject of Multidisciplinary team in HTX patients. The issue is well-presented and a useful clinical conclusion is extracted.
Author Response

(The authors gave the same response as above.)
